# Global Trends in Urban Agriculture Research: A Pathway toward Urban Resilience and Sustainability

**Dan Yan** [1,2], **Litao Liu** [3], **Xiaojie Liu** [3,*] **and Ming Zhang** [2]

1   Center for Energy, Environment & Economy Research, Zhengzhou University, No. 100 Science Avenue, Gaoxin District, Zhengzhou 450001, China; yandan@zzu.edu.cn
2   Tourism Management School, Zhengzhou University, No. 100 Science Avenue, Gaoxin District, Zhengzhou 450001, China; 202012022010531@gs.zzu.edu.cn
3   Institute of Geographic Sciences and Natural Resources Research, Chinese Academy of Sciences, 11A Datun Road, Chaoyang District, Beijing 100101, China; liult@igsnrr.ac.cn
*   Correspondence: Liuxj@igsnrr.ac.cn

**Abstract:** Urban agriculture has been proposed as an important urban element to deal with the challenges of food insecurity and environmental deterioration. In order to track current popular topics and global research trends in urban agriculture, we used bibliometric analysis and visualization mapping to evaluate and analyze the developments in the knowledge of urban agriculture based on 605 papers from the core collection database Web of Science from 2001–2021. The results were as follows. (1) The number of urban agriculture publications increased substantially year by year, indicating that the field is attracting increasing attention. The University of Kassel, Chinese Academy of Sciences, and University of Freiburg are the most productive research institutions in the field of urban agriculture. The top-five most influential countries are the Unites States, Germany, the United Kingdom, Italy, and China, of which the Unites States plays a central role in the cooperative linkage between countries. (2) Research on urban agriculture focuses not only on food production and different styles but also on how to realize the various functions of urban agriculture. In addition, UA-related sustainability and the water-energy-food nexus have become two emerging research topics. (3) Urban agriculture does not necessarily mean a resource-conserving and environmentally friendly food system. To achieve sustainable development, a transition based on technological innovation is needed. How to improve the sustainable development level of the food system while fully considering the resilience, sustainability, and versatility of urban agriculture is the main direction of future research.

**Keywords:** urban agriculture; bibliometrics; CiteSpace; bibliometrix

## 1. Introduction

Urban agriculture (UA) has been practiced for as long as there have been cities [1]. UA was first defined as agriculture producing perishable products, for example, vegetables, animal products, and flowers, in the peri-urban area. As the world is undergoing rapid urbanization, great pressure is being placed on the food supply and urban environment, especially in fast-developing cities [2]. In the process of rapid urbanization, the concept of UA and its associated activities continue to evolve as development requirements and new focuses emerge. UA has diverse forms in terms of function, labor, and management, as well as the integration of these markets. However, it is difficult to map every example of UA neatly onto a single category, due to the fact that there may be overlap [3]. Therefore, harmonization of the definition of UA is lacking. At present, many scholars agree that UA is a form of modern agriculture, emphasizing its multiple functions in ensuring food security, maintaining urban ecosystem services, and improving urban life quality [4–8].

Previous studies have confirmed the multiple socioeconomic and environmental benefits of UA, including contributions to food and nutrition security, livelihood improvement,

ecosystem service provisions, resource conservation, pollution reduction, and urban beautification [9–13]. In recent years, global issues, such as climate change have severely affected food security and food system sustainability [14–17]. UA has been considered as an effective strategy to mitigate climate change, because it could reduce greenhouse gas emissions by shortening the food supply chain and decreasing the food quantity and quality losses caused by long-distance transportation [18]. Since 2019, the COVID-19 pandemic has had an unprecedented impact on food security around the globe. Strict lockdown measures were implemented in many countries to prevent the intracommunity spread of COVID-19. These strict lockdown measures not only prevented the movement of local and migrant workers but also disrupted the supply chain of urban farmers' markets, which placed severe stresses on the food supply of many countries and cities [6]. As an advanced form of agricultural development, UA has become a popular topic in the field of agriculture research. UA, particularly innovative and disruptive solutions, is seen as having the potential to contribute to more resilient and sustainable cities and food systems, and strengthen local food production during and after the COVID-19 pandemic [19–24]. Although the food supply function of UA has been highlighted due to the pandemic, the positive potential of non-material aspects of urban gardens in the creation of therapeutic landscapes in and beyond COVID-19 has been discussed [25]. In addition, UA can be an efficient approach to promote the integrated development of urban and rural areas by breaking geographical and institutional barriers and by promoting cooperation between urban and rural areas in terms of technologies, capital, and talent [26].

Over the past decade, international studies in UA have focused primarily on the development of the concept of UA, agriculture pollution, ecosystem services, nutrient management, urban planning, impact assessment, case studies (such as green spaces in schools, in home gardens, and on rooftops, as well as suburban agriculture and vertical farms), and the role of UA in response to the COVID-19 crisis [15,23–25,27–33]. Diverse UA research topics have been formed because of improvements in theories and systems. Evaluating the research status and popular UA topics, exploring its historical evolution, and conducting a relatively complete analysis of the development of international UA from a macro perspective can provide a theoretical reference for academic research in UA, which will play an important role in promoting the development of UA.

Several previous studies have reviewed the UA literature [31,34–37]. For example, Lwasa, et al. (2015) reviewed the literature on urban and peri-UA and forestry (UPAF) and found emerging consensus on the potential of UPAF in adaptation but found less agreement with respect to the mitigation of climate change [38]. Azunre, et al. (2019) discussed the role of UA in the sustainable city discourse and concluded that UA supports the economic, social, and environmental sustainability of cities [39]. Few studies, however, have applied bibliometric analysis to visualize the qualitative and quantitative changes in UA or related fields, particularly from an interdisciplinary perspective [36]. In addition, few literature reviews have focused on the potential of UA in terms of global issues, such as public health security, food security, and climate change.

To track current popular topics and global research trends in urban agriculture, we used 605 papers published between 2001 and 2021 in the core database of Institute for Scientific Information (ISI) Web of Science (WoS) as the data in this study. The information visualization software CiteSpace and bibliometrix package in the R language were used to map the knowledge structure and development process of UA research. The highly influential countries, institutions, authors, journals, and co-citation network were identified. We also described popular research topics and cutting-edge trends of UA in different periods and assessed the role of UA in achieving a series of global, regional, and national goals (e.g., sustainable development goals (SDGs),carbon peak, and carbon-neutral targets). In addition, we reviewed the importance of UA in improving urban resilience and sustainability as well as its future research trends.

## 2. Methodology

### 2.1. Date Source

The bibliographic data used for this study were obtained from the WoS Core Collection database provided by the ISI, including SCIE, SSCI, AHCI, ESCI, BKCI, and CPCI, on 11 April 2021. We used "urban agriculture" as a search keyword and searched for journal research articles published between 1 January 2000 and 31 March 2021. After excluding meeting minutes, errata letters, and documents with duplication and missing data, we obtained a total of 605 papers.

### 2.2. Methods

Scientific knowledge mapping is a relatively new research method in scientometrics and informetrics. It can not only detect and visualize emerging trends and transient patterns in scientific literature but also identify the knowledge structure, evolution, and development in related fields. Numerous software tools are used to conduct bibliometric analysis, of which CiteSpace is one of the most popular. CiteSpace is a Java application that supports visual exploration with knowledge discovery in bibliographic databases. This tool offers several options to understand and interpret network and historical patterns, such as the growth of a research topic, the main citations in the knowledge base, the automatic labeling of the different clusters using terms from citing articles, geospatial collaboration networks, and international collaboration [40,41]. The bibliometrix R package is an open-source tool used to execute a comprehensive science mapping analysis of scientific literature, and it has three main functions: data collection, data analysis, and data visualization [42]. Because of integration with other statistical and graphical packages in R, it offers great advantages in statistical computing and graphic visualization.

In this study, the CiteSpace and Bibliometrix R package were used to make a quantitative analysis of the relevant literature in the field of UA. We constructed a corresponding knowledge map and identified high-yield authors, international cooperation, high-frequency keywords, co-citation networks, keyword co-occurrence, and historical trajectories. We sorted the overall development context, popular research topics, and core topics of international UA research; these results can provide useful references for future UA research.

## 3. Popular Research Topics and Frontiers in Urban Agriculture

### 3.1. General Information and Annual Publications

The annual number of published papers can reflect the degree of attention to this specific research field to some extent. As shown in Figure 1, a total of 605 papers were published on UA from 2001 to 2021. The number of published papers increased substantially year by year, and the entire period can be divided into three stages.

1.  Slow development (2001–2008): At this stage, UA slowly attracted the attention of scholars. The number of papers published each year was less than 10, but followed a slightly increasing trend.
2.  Steady development (2009–2014): The number of peer-reviewed papers published in the field of UA showed a steady increase during 2009–2014. For each year, the annual number of published papers was about 20–40.
3.  Rapid development (2015–2021): The number of papers published during this period grew rapidly, of which 79 papers were published in 2020, and 23 articles were published in the first quarter of 2021. It is expected that there will be more publications in the future. As a popular emerging topic in agricultural research, UA is gradually attracting the attention of scholars. A large gap still exists, however, between UA and many traditional research topics.

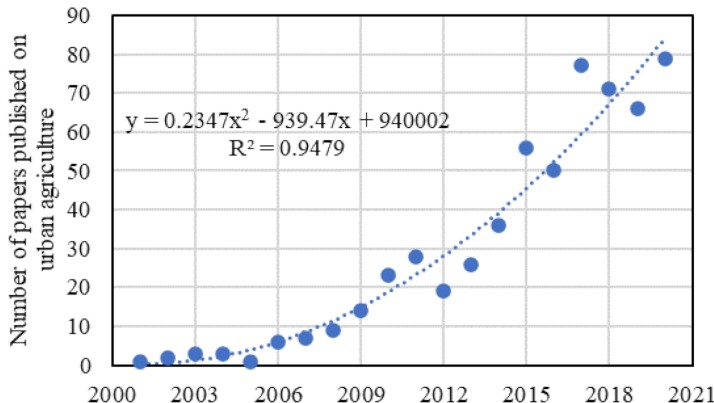

**Figure 1.** Number of papers published on urban agriculture from 2001 to 2020.

*3.2. Active Countries, Institutions, and Authors*

3.2.1. Active Countries

The overall research performance and influence of a country in a particular field can be measured by the number of high-quality papers published and the frequency of citations [43]. Sixty-four countries published papers on UA between 2001 and 2021. The top-six countries are the Unites States, Germany, the United Kingdom, Italy, China, and Canada. Table 1 and Figure 2 show that the United States is the most influential country in this field, and it ranks first in terms of the number of published papers, total frequency of citations, and national cooperation network. Scholars in the United States published 132 papers from 2001 to 2021, accounting for 22% of the total published papers, of which 16 papers are multi-country cooperative papers, covering topics such as food system design, food security, food-energy-water nexus, and case studies in different regions. The total number of citations for these papers is 2372, with an average of 18 citations for each paper. Germany and the United Kingdom rank second and third, respectively, accounting for 6.7 and 5.7% of the world's total published papers in this field. The researchers in Germany focused mainly on the sustainable development of cities, food security, land use, and the socioeconomic and environmental impact assessment of UA, whereas British scholars paid more attention to food security, climate change, agricultural pollution, and urban livelihood. Note that the number of published papers is not always consistent with the total frequency of citations. Although authors from China, Brazil, and South Africa have published a significant number of papers, the frequency of citations was less than 10.

**Table 1.** Top 10 countries with the highest number of papers published on urban agriculture.

| Countries | Number of Papers | Frequency | SCP | MCP | Total Number of Citations | Average Citations for Each Paper |
|---|---|---|---|---|---|---|
| USA | 132 | 22.0 | 116 | 16 | 2372 | 18.0 |
| Germany | 40 | 6.7 | 17 | 23 | 812 | 20.3 |
| UK | 34 | 5.7 | 21 | 13 | 611 | 18.0 |
| Italy | 31 | 5.1 | 21 | 10 | 825 | 26.6 |
| China | 30 | 5.0 | 19 | 11 | 293 | 9.8 |
| Canada | 27 | 4.5 | 21 | 6 | 608 | 22.5 |
| France | 27 | 4.5 | 17 | 10 | 439 | 16.2 |
| Brazil | 25 | 4.2 | 17 | 8 | 68 | 2.7 |
| South Africa | 25 | 4.2 | 19 | 6 | 202 | 8.1 |
| Australia | 24 | 4.1 | 19 | 5 | 500 | 20.8 |

SCP: Single Country Publication; MCP: Multiple Country Publication.

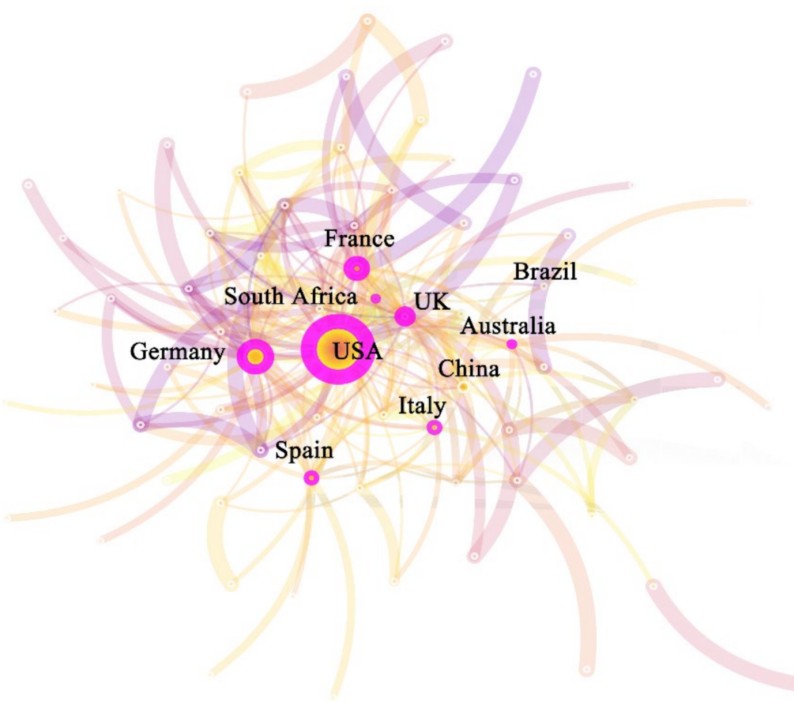

**Figure 2.** Network of national cooperation in the field of urban agriculture from 2001 to 2021.

### 3.2.2. Active Institutions

From 2001 to 2021, 755 institutions were involved in the field of UA research. The top eight high-yield institutions are as follows: University of Kassel, Chinese Academy of Sciences (CAS), University of Freiburg, International Water Management Institute, University of Minnesota, Humboldt University, University of California–Berkeley, and University of Bologna. As shown in Figure 3, the CAS cooperates closely with many institutions. The size of the CAS's annual ring rank is slightly smaller than that of the University of Kassel, but it is bigger than that of the other institutions, which indicates its leading position in this field. The research of the CAS in the field of UA involves many aspects, such as environmental science, rural tourism, agricultural pollution, carbon emissions, land use, and eco-system services.

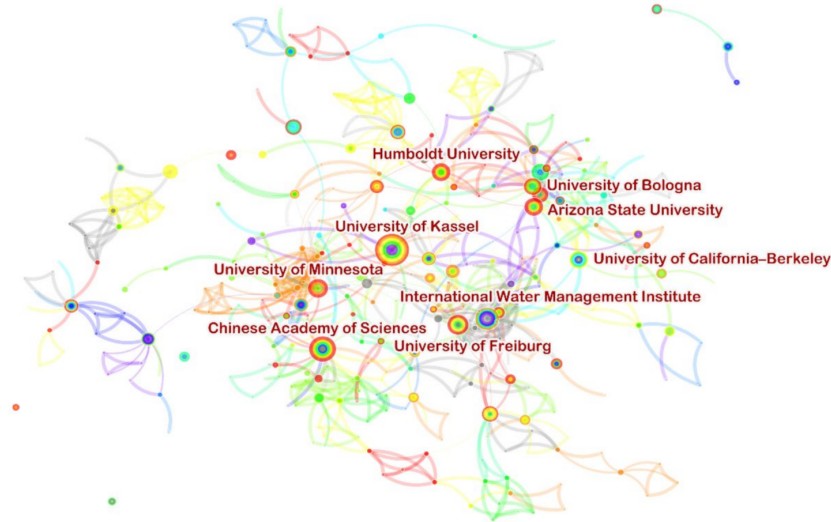

**Figure 3.** The influence of institutional cooperation in the field of urban agriculture from 2001 to 2021.

### 3.2.3. Active Authors

The number of papers published by an author in a certain field and the frequency of citations made by the author can reflect his/her contributions to the field, which can help people identify the leading authors in this field [44]. Figure 4 shows the top 10 authors who have published papers on UA. Most of the top 10 authors have conducted long-term research on UA. For example, Professor Buerkerta from the University of Kassel in Germany has published papers on UA since 2008 and continued publishing until 2019. Among the top 10 authors, there are no Chinese scholars. Although the influence of Chinese scholars and institutions is becoming stronger, few scholars have carried out long-term research on UA and have continued publishing papers in international journals. Difficulty in applying for long-term funding for scientific research on UA is the main reason that research in this field cannot be conducted continuously.

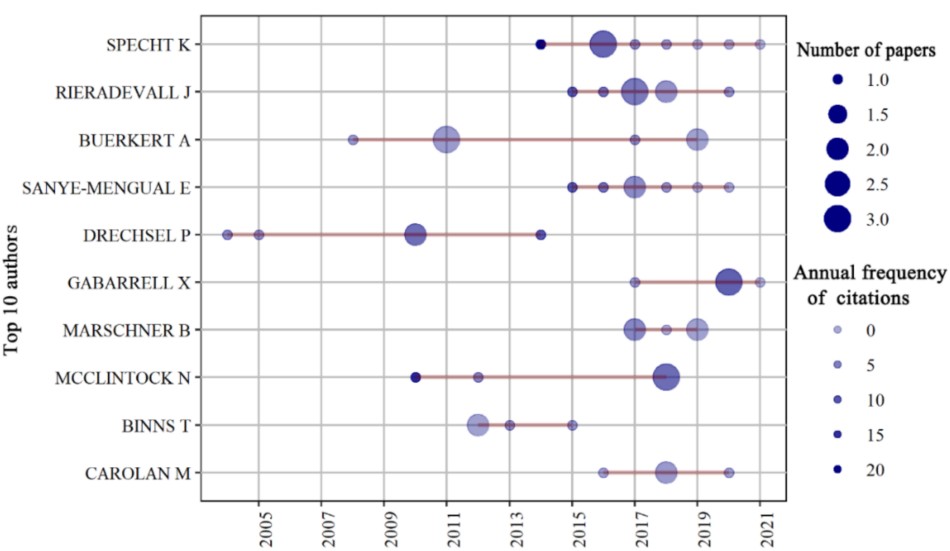

**Figure 4.** Top 10 authors' productivity in the field of urban agriculture from 2001 to 2021.

### 3.2.4. Active Journals

More than 21% of the papers in this field were published in the top 10 active journals listed in Table 2. The top 10 journals are mostly international interdisciplinary journals, focusing on sustainable development, land use, environment, energy, urban planning, agricultural systems, and food systems. Sustainability published 29 papers concerning UA, followed by Land Use Policy and Agriculture and Human Values. Journal of Cleaner Production (IF: 9.297) has the highest impact factor, followed by Science of The Total Environment (IF: 7.963), Landscape and Urban Planning (IF: 6.142), and Land Use Policy (IF: 5.398), while the other journals have a relatively low impact factor. From the perspective of a journal's quartile ranking (JCR), there are six Q1 journals and two Q2 journals, which indicates that the overall quality of journals is high. The h-index values of the top 10 journals are not high, however, and only four journals have an h-index value higher than 100. The average citations of the UA-related papers published in the six Q1 journals range between 15.6 and 34.5, while the average citations of the UA-related papers published in the two Q2 journals are from 5 to 19. It is obvious that the UA-related papers within these journals had a relatively high impact.

**Table 2.** Top 10 journals with the most articles published in the field of urban agriculture from 2001 to 2021.

| Journals | Numbers of Paper | IF | Quartiles | h-Index | Average Citations for Each Paper | Countries |
|---|---|---|---|---|---|---|
| Sustainability | 29 | 3.251 | Q2 | 85 | 5 | Switzerland |
| Land Use Policy | 24 | 5.398 | Q1 | 115 | 24 | Netherlands |
| Agriculture and Human Values | 18 | 3.295 | Q1 | 76 | 34.5 | Netherlands |
| Science of The Total Environment | 13 | 7.963 | Q1 | 244 | 15.9 | Netherlands |
| Journal of Cleaner Production | 12 | 9.297 | Q1 | 200 | 15.6 | USA |
| International Journal of Agricultural Sustainability | 8 | 2.905 | Q1 | 39 | 31.3 | UK |
| Landscape and Urban Planning | 8 | 6.142 | Q1 | 161 | 29.4 | Netherlands |
| Renewable Agriculture and Food Systems | 8 | 2.657 | Q2 | 53 | 19 | UK |
| Future of Food: Journal on Food Agriculture and Society | 7 | — | — | 6 | 2.9 | Germany |
| Cahiers Agricultures | 6 | — | — | 19 | 2.8 | France |

### 3.3. Keywords in Urban Agriculture Research

Keywords are scientific terms that serve as a summary or key to a study that help other researchers locate a paper. Therefore, analyzing the keywords in the field of UA research can identify the popular topics and future trends in this field.

#### 3.3.1. Keywords Network Analysis

The top 10 keywords in the field of UA research are urban agriculture, food security, peri-urban agriculture, urban planning, sustainability, agriculture, urbanization, urban farming, sustainable development, and urban and peri-urban agriculture (Figure 5). Urban agriculture was the most popular keyword and was used a total of 278 times, and its node was the largest, but there were fewer links between "urban agriculture" and other keywords. "Food security" ranked second, and it was closely related to the keyword "urban planning". "Peri-urban agriculture" had a strong connection with two keywords: "city" and "space". "Urbanization" normally appeared simultaneously with keywords such as "urban farming", "benefit", and "community garden." Based on the keyword analysis, we found that popular international research topics on UA included food security, agriculture in peri-urban areas, urban agriculture and urban planning, sustainable urban agriculture, urbanization, and urban agricultural development. In addition, the keyword "Helminth" also attracted attention. Researchers from a different discipline may have trouble understanding this keyword. Helminths are invertebrate animals that comprise a broad spectrum of different pathogens able to affect human health, which in turn contaminate soil in areas where sanitation is poor [45].

#### 3.3.2. Keyword Co-Occurrence Analysis

Keyword co-occurrence refers to the situation in which two or more papers use the same keyword. Keyword co-occurrence can effectively reflect the popular research topics in the discipline and can provide auxiliary support for scientific research [46]. We constructed a UA keyword co-occurrence network using R (Figure 6). Fifty keywords were divided into eight groups based on the keywords' co-occurrence frequency, which was represented by different colors. The number of links between two nodes referred to the frequency of co-occurrence. This number can be used as a quantitative index to depict the relationship between two nodes.

The results showed that "city", "food", and "community garden" were the keywords with the highest frequency of co-occurrence with other keywords, which indicated that UA-related research has focused primarily on urban food production and urban green space. As a type of UA with the highest frequency of co-occurrence, "community garden" has played an important role in building urban resilience by improving food security and public health. The second group of keywords included "food security", "management", "urbanization", "health", "cities", and "ecosystem services", which also were closely related to other groups of keywords. Cites are complex systems. During urbanization, a population gathers intensively in cities. The need for UA to provide insight on the economic, environmental,

and social functions is urgent to support a healthy city life. Therefore, managing UA to fulfill these functions is another popular topic in the field of UA research. In addition, we also found that "carbon", "water", "food production", and other keywords belonging to group 8 were related to "energy" and "future". The above keywords in group 8 were consistent with the current international research topic of "water-food-energy nexus".

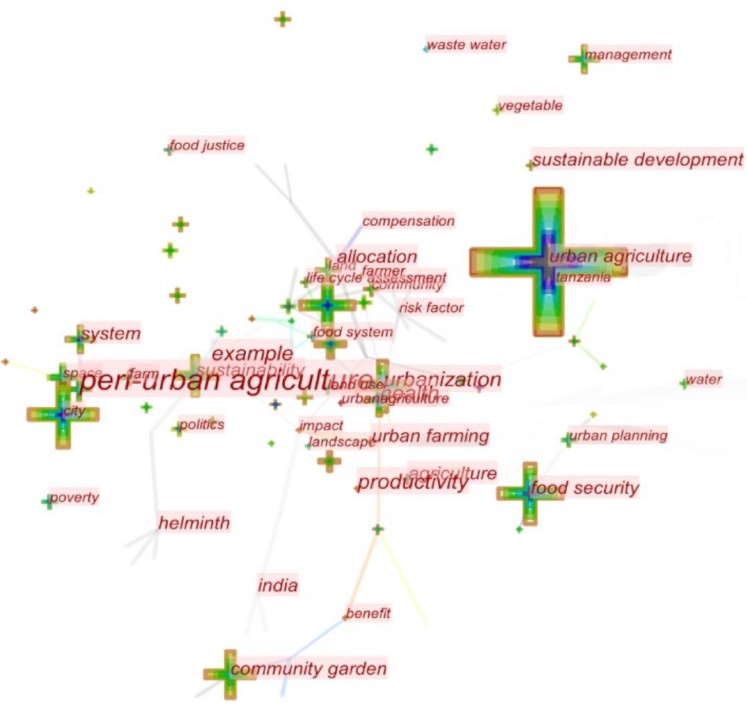

**Figure 5.** Network of keywords in the field of urban agriculture from 2001 to 2021.

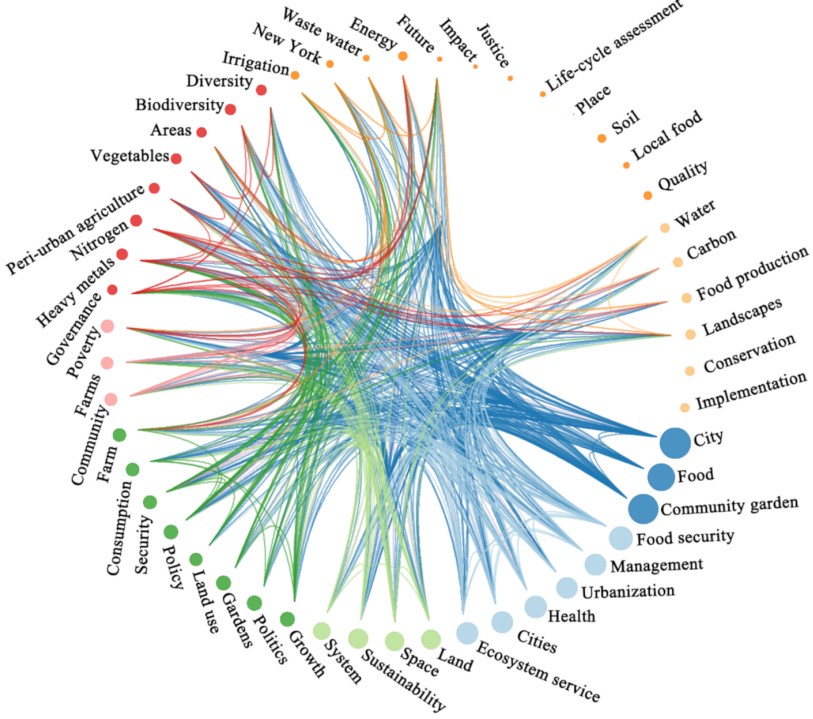

**Figure 6.** Co-occurrence analysis of keywords in the literature on urban agriculture from 2001 to 2021.

*3.4. Cluster Analysis*

Cluster analysis can be used not only to identify important turning points in the evolution of UA research and research frontiers but also to examine the correlation among research frontiers. In this study, we used CiteSpace for mapping and clustering the bibliometric network of UA. The parameter configuration was as follows: the time span was 2001–2021; the time-cut was set to one year; the sources of the topic words included the title, abstract, and author keywords; the topic type was noun phrases; the node type was cited literature; the threshold was the top 50; and the pruning scheme was the minimum spanning tree to trim the time zone segmented network. The results showed that there were 11 clusters in the field of UA research in the period from 2001 to 2021 (Figure 7). Figure 8 charts the timeline of the 11 clusters, showing the time span of the different research topics. According to the time span of these different research topics, we categorized 11 research topics into the following three types: long-term topics, short-term topics, and emerging research topics.

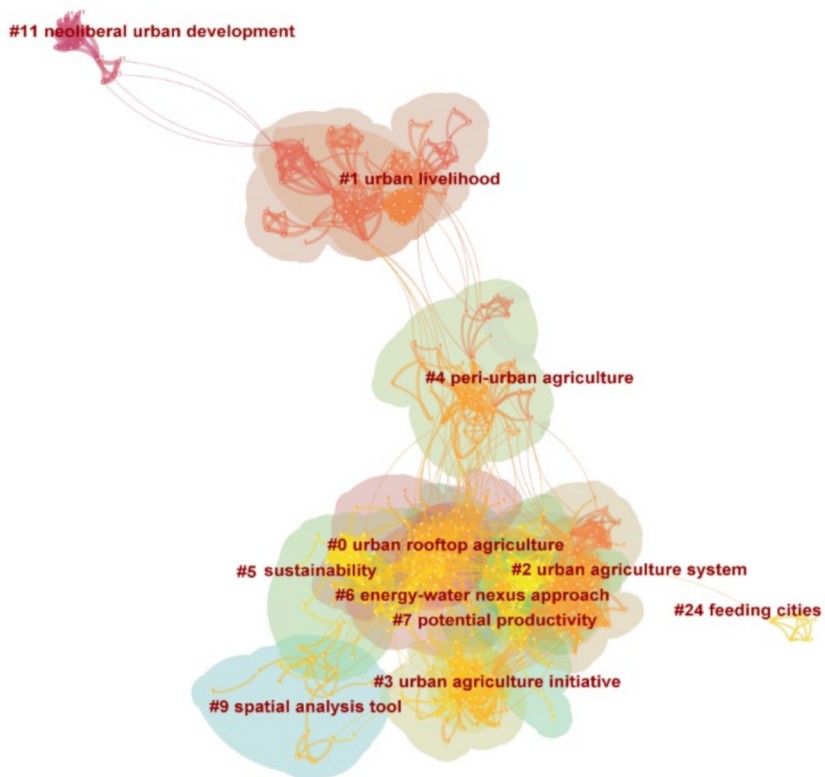

**Figure 7.** Cluster analysis of papers published on urban agriculture from 2001 to 2021.

3.4.1. Long-Term Research Topics

Long-term research topics in UA can last for 10 years or more. Five topics in UA can be considered long-term research, including cluster #0, urban rooftop agriculture; cluster #2, urban agriculture system; cluster #3, urban agriculture initiative; cluster #4 peri-urban agriculture; and cluster #7, potential productivity.

**Cluster #0: Urban rooftop agriculture.** This cluster included 83 papers and represented extensive concerns about rooftop agriculture. Rooftop agriculture is a building-based form of UA, which is considered to be a useful strategy for targeting global concerns, for example, food security, limited land resources, and climate change [35]. Researchers examined a range of aspects, including the evaluation of the social acceptance of rooftop agriculture, potential of rooftop agriculture, impact assessment of rooftop agriculture on the environment, sustainable development of rooftop agriculture, and rooftop agriculture management [47–55]. For example, Specht, et al. [10] used the framework of sustainabil-

ity to understand the role of rooftop agriculture in future urban food production, and reviewing its major benefits and limitations, and found that rooftop agriculture is not sustainable by itself and needs to be managed properly [10]. Goldstein, et al. (2016) used an environmental life-cycle assessment for six urban farms in the United States producing lettuce and tomatoes and concluded that UA may not be the optimal application of space in northern cities to improve urban environmental performance [56]. Orsini, et al. (2014) estimated the contribution of rooftop vegetable production to food security in Bologna and found that rooftop agriculture could provide more than 12,000 t year$^{-1}$ vegetables to Bologna, satisfying 77% of the inhabitants' needs [53].

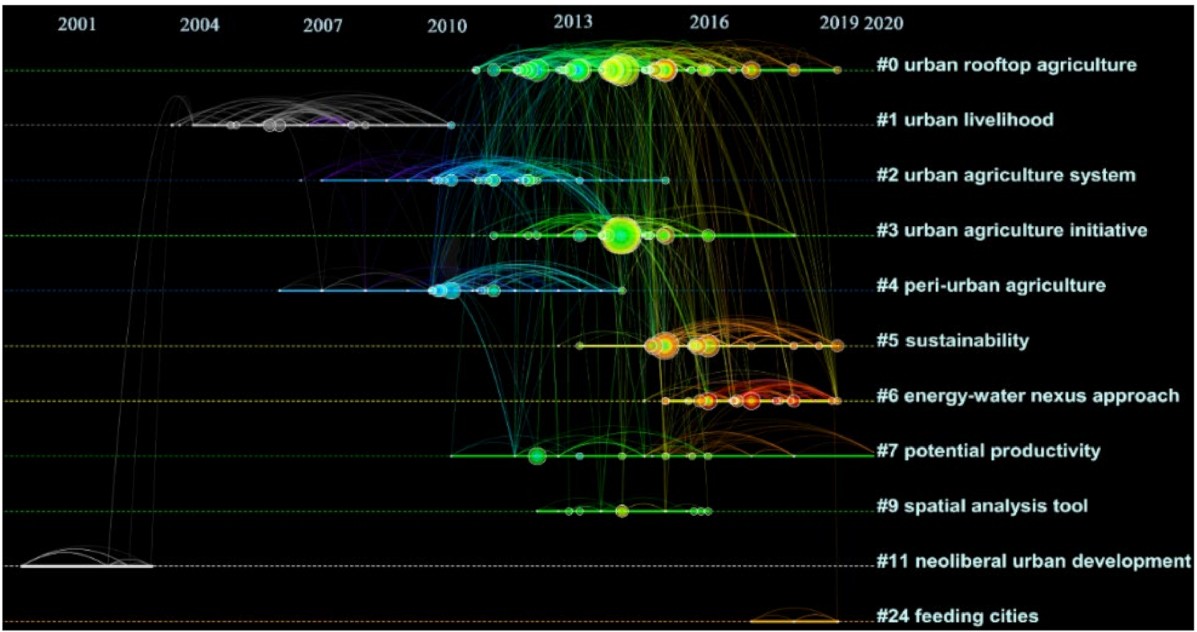

**Figure 8.** Timeline of clusters on urban agriculture from 2001 to 2021.

**Cluster #2: Urban agriculture system.** This cluster included 59 papers. The research focused on case studies of different urban agricultural systems, assessment of environmental impact of urban agricultural systems, policy making, and sustainable development of the urban agriculture system [7,57–59]. For example, Cohen and Reynolds (2015) conducted a two-year study in New York City and suggested that the significant disparities in access to resources make the urban agriculture system in New York unequal and constrain the efforts of some farms and gardens [60]. Martellozzo, et al. (2014) assessed the self-sufficiency of vegetable production in Montréal under different land-use scenarios and concluded that UA could substantially increase the self-reliance and resilience of food systems in North American cities [28].

**Cluster #3: Urban agriculture initiative.** This cluster included 46 papers. The core idea of this cluster is to propose and implement an effective UA development plan to achieve such targets as poverty reduction, food security, waste recycling, job creation, income growth, climate change adaptation and mitigation, environmental protection, resource management, ecological landscape restoration, and social equity. For instance, White and Bunn (2017) investigated four UA projects in the UK and found that negotiating access to land and securing long-term funding are common challenges for local UA practitioners. They also identified two pathways for UA in Scotland: partnerships between activists and local government and enabling national legislation to harness local urban food growing and support UA [61]. Frayne, et al. (2014) analyzed data generated by the African Urban Food Security Network's (AFSUN) baseline food security survey undertaken in 11 Southern African cities and found that the economic, political, and historical circumstances and conditions of a city were key factors that could either promote or hinder UA activity and

scale. Wealthier households derive greater net food security benefits from UA than poor households [62].

**Cluster #4: Peri-urban agriculture.** This cluster included 43 papers that covered a wide range of topics. In fact, it is not possible to completely distinguish UA from peri-urban agriculture. Many papers adopted a more concise way of expressing UA. Zezza and Tasciotti (2010) used household survey data from 15 developing countries to analyze the importance of peri-urban agriculture for the urban poor and food insecure. They concluded that the role of UA in urban poverty and food insecurity reduction should not be overemphasized, as its share in income and overall agricultural production is often quite limited. However, its role should not be underestimated, especially in some African regions where agriculture is the main source of household livelihood [63].

**Cluster #7: Potential productivity.** This cluster included 24 papers that focused mainly on how to evaluate and improve the potential productivity of UA to ensure urban food security. For example, Benis, et al. (2018) used cost–benefit analysis to compare alternative productive uses of rooftops in the Mediterranean climate and found that food production is more beneficial than energy generation [64]. Taylor (2020) evaluated the long-term, relative performance of four different systems of small-scale vegetable production appropriate for peri-UA in the US state of Rhode Island. The results indicated that all four systems could be highly productive, with varying trade-offs in terms of their sustainability and impacts on soil quality [65].

3.4.2. Short-Term Research Topics

Research areas in this group were discussed actively for a short period of time and then became less popular. Four topics can be considered short-term research topics, including cluster #1, urban livelihood; cluster #9, spatial analysis tool; cluster #11, neoliberal urban development; and cluster #24, feeding cities. Cluster #1, urban livelihood, had the greatest number of papers.

**Cluster #1: Urban livelihood.** Cluster #1 included 68 papers. The studies in this cluster focused mainly on the effect of UA on urban livelihood improvement. For example, Simatele, et al. (2012) explored the relationships between urban livelihoods and extreme weather events and evaluated the extent to which changes in climate and urban governance affected UA, based on field-based research in Lusaka, Zambia [66]. Olivier (2019) applied a sustainable livelihood framework to a case study on cultivators from Cape Town's Cape Flats and found that cultivators use UA in highly complex ways to build sustainable livelihoods. Non-governmental organizations are central to this process, and distrust, crime, and a lack of resources are limiting factors [67]. Adeoti, et al. (2011) examined the sustainability of livelihoods through UA and confirmed the high contribution of UA to urban farmers' livelihoods. They found that farm size and access to credit were significant determinants of income from UA [68].

**Cluster #9: Spatial analysis tool.** Cluster #9 included 17 papers. It mainly focused on using remote sensing and geographic information system (GIS) technologies to provide a framework for spatial analysis and modeling based on geographic principles. Using the technologies was an attempt to integrate the analytical capabilities to broaden the understanding of the real-world system [69,70]. For example, Thapa and Murayama (2008) presented an integrated technique of an analytical hierarchical process and GIS to evaluate the land for peri-urban agriculture in Hanoi province, Vietnam, and classified the land resources in Hanoi into four groups: high-suitability, medium-suitability, low-suitability, and unsuitable land for peri-urban agriculture [71]. Pulighe and Lupia (2016) proposed a conceptual framework and methodology for mapping the UA in Rome using Earth Observation techniques. They argued that the integration of the web-mapping services for building urban agricultural land-use datasets was cost-effective compared with processing commercial remote sensing images [72].

### 3.4.3. Emerging Research Topics

Two popular topics are emerging in the field of UA research: cluster #5, "sustainability", which has attracted much attention since the United Nations proposed its 17 SDGs; and cluster #6, the "energy–water nexus approach", which keeps pace with the international research hotspot, water-energy-food nexus.

**Cluster #5: Sustainability.** Cluster #5 included 34 papers. Few studies, however, focused on the sustainability of UA itself [73]. For example, McDougall, et al. (2019) investigated 13 urban community gardens and concluded that UA can be a highly productive use of land, with each square meter put under cultivation considered equivalent to nearly twice that area of rural farmland [74]. UA also requires the judicious management of inputs to achieve sustainability. The majority of previous studies focused on how UA could improve the sustainability of cities, energy, the environment, and food systems [29,75–78]. For instance, Langemeyer, et al. (2021) discussed the importance of UA in promoting food resilience, global sustainability, and multifunctionality. They argued that existing and new knowledge about urban risks and vulnerabilities, the spatially explicit urban metabolism (e.g., energy, water, nutrients), and ecosystem services need to be stronger and should be jointly considered in land-use decision making [6].

**Cluster #6: Energy–water nexus approach.** Cluster #6 included 29 papers. In recent years, the water-energy-food (WEF) nexus has become a popular research topic in the field of sustainable development because of its potential to help understand trade-offs and synergies among WEF interactions [79]. As a primary production process, UA can be considered a component embedded in the urban food system, which closely interacts with urban resource streams, such as water and energy [80]. Therefore, it is of great importance to apply the WEF nexus approach in UA. Caputo, et al. (2020) carried out an energy-environmental assessment for peri-urban agriculture in Milan, Italy, and found that the different practices used in peri-urban agriculture can make a positive contribution toward achieving the Agenda 2030 goal of ensuring sustainable farm production practices [81]. Subsequently, Caputo, et al. (2021) proposed a conceptual basis for a UA nexus, along with an assessment methodology that explicitly included social dimensions in addition to food, energy, and water [82].

## 4. Discussion

As mentioned in Section 3.4, several researchers found that UA could substantially increase the self-reliance and resilience of food systems in cities. For example, Martellozzo, et al. (2014) suggested that UA could meet the world's vegetable needs with one-third of the urban area, based on current yields [28]. Orsini, et al. (2014) found that rooftop agriculture can provide 77% of the urban vegetable requirements in Bologna, Italy. New evidence was recently provided in support of this opinion [53]. According to a report for modern UA development in China released in February 2021, China's 35 largest cities can supply 76% of the vegetable consumption of 330 million residents, with arable land accounting for 30% of the urban planning area. In Shanghai, UA can satisfy up to 90% of local vegetable needs. Some researchers, however, hold a pessimistic view of the potential of UA. For example, McClintock, et al. (2013) assessed the potential contribution of vacant land to urban vegetable production and consumption in Oakland, California. Results showed that UA could contribute only 2.9% to 7.3% of Oakland's consumption, depending on production methods [83]. Although there are still doubts about the yield and food supply capacity of UA, most scholars agree that UA plays an important role in enhancing urban food security.

Despite the fact that the importance of UA has been noticed by the world, UA has not received considerable attention in academic circles. Taking annual number of published papers in the core database of Institute for Scientific Information (ISI) Web of Science (WoS) as an example, only 79 UA-related papers were published in 2020, accounting for 0.46% of the agricultural papers published in the same year. There are very few specialized journals on urban agriculture. The top 10 journals are mostly international interdisciplinary journals,

focusing on sustainable development, land use, the environment, energy, urban planning, agricultural systems, and food systems. Even for the top 10 most influential countries, many have produced little UA-related research. For example, Chinese scholars have published numerous papers in international journals, but the total frequency of citations is not high. Few scholars in China have carried out long-term research on UA but continue to publish papers in international journals. This is certainly related to the lack of long-term funding for UA research, but it also represents a lack of interest in UA.

The COVID-19 pandemic has posed a tremendous global challenge and has impacted every aspect of human life. A large number of studies on COVID-19 have been published in various scientific fields instead of public health security and medical research. In terms of food security, the COVID-19 pandemic not only prevented the movement of local and migrant workers but also disrupted the supply chain of urban farmers' markets, which placed severe stresses on the food supply of many countries and cities [6]. As an effective measure to cope with the COVID-19 pandemic, UA should have attracted the attention of scholars during the pandemic. However, only a few scholars have conducted COVID-19-related research in the field of UA. Until 11 April 2021, only four papers, which discussed the impact of the COVID-19 pandemic on UA and the significance of UA in promoting urban resilience during the pandemic period, were published in the core database of Institute for Scientific Information (ISI) Web of Science (WoS). It is apparent that UA-related research has not responded quickly to the emerging global crisis.

Previous studies suggested that improving the resilience of food production and supply will become particularly important for coping with multiple shocks, such as public health security, rapid population growth, and urbanization. The challenge is to harness the lessons of different shocks to promote more resilient urban food systems. Compared with more traditional rural agriculture, UA is considered to be more resilient because of its short food supply chain and diversified production systems [84]. In the process of planning urban food systems and developing global sustainable development strategies, policy makers should fully consider the resilience, sustainability, and versatility of UA. Future research directions in the field of UA may include economic, social, and environmental impact assessments as well as how to improve the resilience and sustainability of UA.

## 5. Conclusions

In this study, bibliometric analysis and visualization mapping were used to evaluate developments in the knowledge of urban agriculture based on 605 papers from the core collection database Web of Science from 2001–2021. The results show that the number of urban agriculture publications increased substantially year by year, indicating that the field is attracting increasing attention. More than 21% of the papers in this field were published in the top 10 active journals, which are mostly international interdisciplinary journals focusing on sustainable development, land use, environment, energy, urban planning, agricultural systems, and food systems. The top five most influential countries are the Unites States, Germany, the United Kingdom, Italy, and China, of which the United States plays a central role in the cooperative linkage between countries. The University of Kassel, Chinese Academy of Sciences, and University of Freiburg are the most influential research institutions in the field of urban agriculture.

Results also show that UA-related research has focused primarily on urban food production and urban green space. In recent years, the need for UA to provide insight on economic, environmental, and social functions is urgent to support healthy cities. Therefore, managing UA to fulfill these functions is another popular topic in the field of UA research. Apart from the above topics, UA-related sustainability and the water-energy-food nexus are two emerging topics consistent with the two popular international research topics. In addition, this study also found that there are still doubts about the yield and food supply capacity of UA, but most scholars agree that UA plays an important role in enhancing urban food security. Urban agriculture does not necessarily mean a resource-conserving and environmentally friendly food system. To achieve sustainable development, a transition

based on technological innovation is needed. How to improve the sustainable development levels of the food system while fully considering the resilience, sustainability, and versatility of urban agriculture is the main direction of future research.

**Author Contributions:** Conceptualization, D.Y., X.L. and L.L.; methodology, D.Y. and X.L.; soft-ware, D.Y.; validation, D.Y. and X.L.; formal analysis, D.Y.; investigation, D.Y., L.L. and M.Z.; re-sources, D.Y.; data curation, D.Y.; writing—original draft preparation, D.Y., X.L. and L.L.; writing—review and editing, D.Y., X.L., L.L. and M.Z.; visualization, D.Y., supervision, X.L.; projection administration, X.L.; funding acquisition, X.L. and D.Y. All authors have read and agreed to the published version of the manuscript.

**Funding:** This research was funded by [the international partnership program of the Chinese Academy of Sciences] grant number [132C35KYSB20200007] and [Soft Science Research Project of Henan Province] grand number [212400410077].

**Data Availability Statement:** The data used in this study can be downloaded at https://www.webofknowledge.com (accessed on 10 January 2022).

**Conflicts of Interest:** The authors declare no conflict of interest.

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
