# Peer review of "Global Trends in Urban Agriculture Research: A Pathway toward Urban Resilience and Sustainability"

_land, doi:10.3390/land11010117_

Round 1

Reviewer 1 Report

Please better organize the paper and avoid so many sub-sub-sections. The cluster names could be written in bold and given as the first sentence of a paragraph. 

The discussion should be separated from conclusions and conclusions should contain policy/research implications. 

Reviewer 2 Report

Thank you for letting me review your article. I really enjoyed it and think it will make a valuable contribution to the literature. However, I have some concerns that need to be addressed before it can be published (please find attached my feedback). My major concern is the discussion needs to be redeveloped and the reference list needs to be made consistent for publication aligning to Lands author guidelines.  

Round 2

Reviewer 2 Report

I really enjoyed reading the revisions. The team has done a very good at making the amendments. There are some minor point I need some clarification on:

1) The data collection took 1 day. This seems very short. Is this the finish point? If so put a date range rather than 11 of April.

2) Figure 3: please put universities in full rather than short hand.

3) Line 230 citations and impact factor are different things so please do not conflate that these paper made a greater impact as they use different formulars to measure them.

4) Figure 5: '"Helminth" is not a term many if any know. Can you please either explain the meaning or change the word to a more universal one. 
